# Advanced Endoscopy for Benign Esophageal Disease: A Review Focused on Non-Erosive Reflux Disease and Eosinophilic Esophagitis

**DOI:** 10.3390/healthcare10112183

**Published:** 2022-10-31

**Authors:** Kenichi Goda, Keiichiro Abe, Akira Kanamori, Masayuki Kondo, Shunsuke Kojimahara, Mimari Kanazawa, Takanao Tanaka, Kazunori Nagashima, Tsunehiro Suzuki, Akira Yamamiya, Koki Hoshi, Keiichi Tominaga, Yuichi Majima, Makoto Iijima, Atsushi Irisawa

**Affiliations:** Department of Gastroenterology, School of Medicine, Dokkyo Medical University, 880 Kitakobayashi Mibu, Tochigi 321-0293, Japan

**Keywords:** esophagus, non-erosive reflux disease (NERD), gastroesophageal reflux disease (GERD), eosinophilic esophagitis, endoscopic diagnosis, narrow-band imaging, image enhanced endoscopy, magnification endoscopy, endocytoscopy

## Abstract

Advanced endoscopy (AVE) techniques include image-enhanced endoscopy methods, such as narrow-band imaging (NBI), and types of microscopic endoscopy, such as endocytoscopy. In the esophagus, AVE first showed diagnostic utility in the diagnosis of superficial esophageal cancer and was then applied to inflammatory disease. This review focuses on non-erosive reflux disease (NERD) and eosinophilic esophagitis (EoE), which sometimes show no abnormal findings on standard white light endoscopy alone. Studies have demonstrated that advanced endoscopy, including NBI magnification endoscopy and endocytoscopy, improved the diagnostic performance of white-light endoscopy alone for NERD and EoE. In this review, we explain why advanced endoscopy is needed for the diagnosis of these esophageal inflammatory diseases, summarize the study results, and discuss future perspectives.

## 1. Introduction

Advanced endoscopy (AVE) includes image-enhanced endoscopy methods, such as narrow-band imaging (NBI) [1], and types of microscopic endoscopy, such as endocytoscopy. Having been in clinical use since the early 2000s, AVE has rapidly come into widespread use over last 20+ years. Unmagnified NBI and other types of AVE improve the early detection rate of head and neck and esophageal squamous cell carcinoma. Furthermore, when combined with magnifying endoscopy, AVE may be used to improve the accuracy of qualitative diagnoses of early esophageal, gastric, and colorectal cancers [2,3,4]. Based on these results, the latest consensus statements and guidelines regarding gastrointestinal cancer now recommend the combined use of standard white-light endoscopy (WLE) and NBI when performing endoscopic screening and surveillance [5,6]. Nevertheless, few clinical studies have been conducted on the usefulness of AVE in cases of non-cancerous benign diseases, and a lack of consensus exists as a result. 

Herein, we presented a review of the clinical significance of the use of AVE, particularly NBI, in the endoscopic diagnosis of benign esophageal disease, non-erosive reflux disease (NERD), and eosinophilic esophagitis (EoE), which have recently had an increase in prevalence. Additionally, it is still difficult to diagnose NERD and EoE endoscopically because there are sometimes no clear findings with WLE alone, as mentioned below. 

We explored and selected previous studies that will be suitable for this review article using PubMed with the keywords “non-erosive reflux disease (NERD)”, “eosinophilic esophagitis (EoE)”, “endoscopic diagnosis”, “image-enhanced endoscopy”, “magnification endoscopy”, and “advanced endoscopy”. 

## 2. NERD

Gastroesophageal reflux disease (GERD) includes two types: erosive GERD, in which reflux disease is accompanied by erosion known as a mucosal break that is visible macroscopically; and non-erosive GERD or NERD, in which reflux disease is not accompanied by a mucosal break. In Japan, the incidence of reflux esophagitis is approximately 10%, and 80% of patients with NERD experience GERD symptoms at least twice per week [7]. Studies conducted in the United States and Europe have also reported that patients with erosive GERD for whom a highly specific endoscopic diagnosis is possible account for between 30% and 40% of all GERD patients, that it is difficult to perform endoscopic diagnoses in NERD patients (who account for over 60% of all patients), and that the sensitivity of WLE is low [8]. Compared to the diagnosis of erosive GERD, for which a mucosal break is used as the standard, the diagnosis of NERD involves a standard known as “minimal change” and includes changes such as erythema without sharp demarcation, white turbidity, and invisibility of vessels. As this standard includes a large number of subjective factors, the concordance rate of endoscopic diagnosis is low [9]. It has been reported that even when using 24-hour esophageal monitoring, a gold standard for diagnosing GERD, one-third of NERD patients show no abnormalities [10]. Although NERD is the major subtype of GERD, there are no gold standards for its diagnosis when performing esophagogastroduodenoscopy (EGD) or other types of tests, which, therefore, requires attention. EGD using WLE is the simplest and most widely used test for the diagnosis of GERD. Therefore, in recent years, the combined use of WLE and NBI has been used to search for an AVE that can be utilized to diagnose NERD with a high degree of accuracy. Currently, the results of investigations employing tests such as Lugol chromoendoscopy, autofluorescence imaging (AFI), and NBI have been reported. 

In one study, Lugol chromoendoscopy was performed on 39 patients with GERD in whom a mucosal break was not detected when WLE was used, showing that 19 patients (49%) had Lugol’s unstained streaks. This suggested the possibility of inflammatory changes in the mucosa [11]. Additionally, typical pathological findings related to reflux esophagitis were observed more frequently in unstained areas than in stained areas. The relationship between this finding and GERD was confirmed in patients whose symptoms improved after the administration of antacids [12]. Although Lugol chromoendoscopy is useful for detecting inflamed mucosa that cannot be detected by WLE in patients with NERD, Lugol solutions have disadvantages such as chest pain, chest discomfort, and allergic reactions. 

In a small case series where the esophagogastric junction was observed using WLE and AFI, the esophageal mucosa was normal when using WLE and green when using AFI. Furthermore, Grade M (white turbidity) mucosa was pink when using AFI [13]. In another case series, when NERD patients were observed using AFI, purple vertical lines appeared in their lower esophagitis, making this technique useful in differentiating NERD from functional heartburn [14]. However, the AFI endoscopy system is not commercially available because it has gone out of production.

The most widely used AVE technology is NBI. NBI uses spectroscopic technology to compose images of short-wavelength blue and green light, for which hemoglobin has a high absorption rate. As the wavelengths are shorter than standard white light, they are absorbed by hemoglobin more highly, which allows this technique to make visible the microsurface pattern and microvascular pattern of the mucosa more clearly [15]. Therefore, when used in combination with magnification endoscopy (ME), NBI can detect minuscule mucosal breaks (microerosion) and abnormal microvessels (intrapapillary capillary loops: IPCLs) on the mucosal surface of the lower esophagus, even in cases that have been previously diagnosed as normal using WLE. 

Approximately 6% of patients diagnosed with Grade M disease according to the Modified Los Angeles Classification have mucosal breaks when tested using NBI with magnification endoscopy (NBI-ME), which in turn leads to their diagnoses being upgraded to Grade A [16]. Grade M is a revised Los Angeles classification. Endoscopic findings of Grade M include minimal mucosal changes of white turbidity or erythema without sharp demarcation (i.e., without erosion) in patients with GERD that correspond to NERD.

In one multivariate analysis performed on NBI-ME images of the esophagogastric junctions of 20 NERD patients and 30 control patients, when either an increased number of IPCLs or dilated IPCLs were used as the endoscopic predictors for NERD and when at the highest odds ratios (7.9 and 9.2%, respectively), the sensitivity and specificity for the combination (increased or dilated IPCLs) were both high, at 80% and 100%, respectively [17]. 

Comparisons of the three groups, including erosive GERD, NERD, and the control, showed that when NBI-ME was utilized, microerosions and increased vascularity that could not be seen in WLE had a clearly higher detection rate in NERD patients than in the control group (52.8% vs. 23.3% and 91.7% vs. 36.7%, respectively). NBI-ME was also useful for the differential diagnosis between the NERD and control groups for the disappearance of two findings: increased vascularity and round pit pattern (sensitivity, 86.1%; specificity, 83.3%). The inter-observer agreement rate for all three findings ranged from good to almost perfect and benign (microerosion (κ = 0.89); increased vascularity (κ= 0.95); and pit pattern (κ = 0.59)) [18]. A recent international, multicenter, randomized controlled trial reported findings suggesting that the ridge-villous pattern seen on NBI images may also be useful as a surrogate marker for NERD [19]. 

Conversely, it has also been reported that histopathological testing is superior to NBI [20]. Nevertheless, the specificity of NBI-ME for NERD patients with histological GERD findings based on vascularity, microerosions, and a combination of vascularity and microerosions was high (specificity: 86.7%, 93.3%, and 98.3%, respectively), indicating the usefulness of NBI-ME [21]. Therefore, it is now believed that invasive biopsy sampling of tissues should be avoided whenever possible in cases of GERD, including cases of NERD that are benign and non-tumorous. 

Very few studies have attempted to apply AVE techniques other than NBI to NERD diagnoses. As mentioned above, the i-scan can also reveal findings such as vascularity and microerosion with a high degree of specificity [21]. In one cohort study, the i-scan was found to have a higher detection rate for minimal changes in dyspeptic patients with GERD than in dyspeptic and control patients [22]. 

Although it is believed that flexible spectral imaging color enhancement (FICE, a digital technology similar to NBI that allows for enhanced visibility of microstructures of vessels and mucosal structures) may identify the micromucosal changes of NERD, the correlation between such microchanges detected by FICE and the GERD symptoms remains unknown [23]. The investigation of linked color imaging (LCI) has shown that when WLE is utilized in combination with LCI, there is a significant improvement in the detection rate of Grade M NERD over that of WLE alone (WLE with LCI: 43/88, 48.9% vs. WLE alone: 29/88, 33.0%; *p* < 0.001). Furthermore, the inter-observer variability also significantly improved with the combined use of LCI (WLE with LCI: 0.856-0.920, WLE alone: 0.375-0.705; *p* < 0.001) [24]. An investigation of blue laser imaging (BLI) has shown that BLI-bright has poor interobserver agreement for minimal changes in NERD, which is also the case with LCI and WLE (0.35/0.43, 0.42/0.51, and 0.42/0.48 for WLE, BLI-bright, and LCI, respectively) [25]. No studies have investigated the diagnostic accuracy of BLI for NERD. We encountered a case of NERD that showed the clinical utility of LCI and BLI magnification endoscopy (Figure 1).

When confocal laser endomicroscopy (CLE), a type of microscopic endoscopy, is utilized, it is possible to reveal an increased number of IPCLs and dilated intercellular spaces in NERD patients, which indicates its diagnostic usefulness [26]. 

Many studies suggested the usefulness of AVE (including NBI) and microscopic endoscopy in the diagnosis of NERD, the most frequently observed type of GERD. Japanese guidelines also indicate that image enhancement and magnification endoscopy are useful for GERD diagnosis [27]. 

## 3. Eosinophilic Esophagitis

EoE is a disease that causes symptoms such as tightness of the chest and dysphagia due to strictures associated with esophageal motility disorder, along with esophageal fibrosis caused by chronic inflammation characterized by massive eosinophil infiltration in the intraepithelial tissue of the esophagus [28]. 

The main cause is believed to be an excessive immune response to antigens ingested during meals and pollen that are inadvertently swallowed. There has been a rapid increase in cases in the United States and Europe, whereas in Japan, it remains a rare disease, with recent reports showing that its incidence is increasing slightly; the incidence among patients who underwent EGD was 0.02% in 2011, which increased to 0.34% in 2018 [29]. The required and reference items used as diagnostic standards are listed in Table 1.

EGD is the most important test, as it allows a definitive diagnosis to be made using endoscopic biopsy. The characteristics of EGD based on the WLE findings for EoE were reported to be: (1) white plaque (exudate), (2) concentric constriction rings (rings), (3) mucosal edema (edema), (4) linear furrows (furrows), and (5) stenosis (stricture), namely, EREFS.

Exudate, or white plaques, cause microabscesses due to infiltration of eosinophils, and concentric rings are furrow-like changes found along the short axis of the esophagus with a width that is wider than the so-called tatami creases. Concentric rings and strictures are believed to occur as a result of fibrosis of the lamina propria of the mucous membrane and layers deeper than the submucosal layer. 

The frequency of the above endoscopic features in EoE patients are as follows: “furrows” are most common at 52%, followed in descending order by “exudate” and “rings”. Inflammation and edema in the mucosa are aggravating factors for “furrows”, which differs from the linear erosion seen in reflux esophagitis. They are characterized by two to three shallow, short folds that converge to form a wedge shape and run vertically. The frequency of the characteristic endoscopic findings for these types of EoE is similar in Asia (including Japan). However, in Asia, the frequency of clinically severe cases in which “stricture”, narrowing, and other types of impaction are caused is lower than in the United States and Europe, as they are rarely seen [30]. 

A meta-analysis of prospective studies found at least one of these characteristic findings in over 90% of EoE patients who underwent EGD [31]. In the case of linear furrows, which are most commonly observed, the frequency of histological eosinophil infiltration was significantly higher in the valley regions of the linear furrows than in the mucosa on the adjacent ridge, indicating that biopsies should be performed initially in the valley regions [32]. One method used to increase the accuracy of biopsy diagnoses in the United States and Europe is EREFS, which is used as an objective index of endoscopy findings [33] (Table 2). 

However, there are many EoE patients who report only mild subjective symptoms and whose routine EGD, the same as WLE, shows no endoscopic abnormalities (7–32%) [31,34,35]. In addition, the esophageal mucosa of EoE is extremely fragile, which makes mucosal peeling over a wide area necessary when a biopsy is performed. Therefore, even in these cases, AVE, such as NBI and microscopic endoscopy, would be promising because studies have shown that the use of NBI requires a smaller number of biopsies to be performed while simultaneously improving diagnostic accuracy [36,37]. 

As no improvement in diagnostic accuracy is achieved in WLE diagnoses when using NBI with non-magnification imaging, only the combined use of NBI and ME (NBI-ME) was the subject of this study. Tanaka et al. and Ichiya et al. reported three important findings: (1) beige-colored mucosa; (2) increased and dot-shaped congested IPCLs; and (3) the absence of cyan vessels (invisibility of submucosal vessels) [36,37]. Ichiya et al. reported that WLE findings based on the five characteristic findings (EREFS), as well as the three NBI-ME findings, were all observed at significantly higher frequencies in their pathology group, including EoE patients, than in their control group. Moreover, the results of an additional investigation when the control group was further subdivided into a GERD group and a non-GERD group showed that of the three NBI-ME findings, there was extremely high sensitivity (88%) and specificity (89–94%) for the absence of cyan vessels. Additionally, negative predictive values also showed extremely high figures for both sensitivity and specificity (84–93%), whereas the positive predictive values showed the highest (87–94%). The inter-observer and intra-observer agreement between the eight endoscopists, including four non-experts, regarding the three NBI-ME findings were 0.83–0.85 and 0.93–0.96, respectively, indicating nearly perfect results in both cases. These results indicate that NBI-ME offers high validity (accuracy) and reliability (reproducibility) when used to diagnose EoE, which, in turn, suggests that an NBI-ME-directed biopsy is useful when making differential diagnoses. 

LCI is an image-enhanced technology that has been the subject of a case study involving EoE [38]. We look forward to validating the usefulness of LCI through either a case series or a control study. 

Recently, we reported the use of endocytoscopy (a type of microscopic endoscopy) for the observation of eosinophils that have infiltrated esophageal squamous intraepithelial tissue collected via direct biopsy [39]. Thus, an endocytoscopy-directed biopsy is suggested as a helpful diagnostic tool for EoE, which requires only a minimum number of biopsy samples. 

We encountered a case of EoE in which NBI-ME was helpful in the diagnosis, and endocytoscopy could visualize infiltrating eosinophils directly in vivo during upper GI endoscopy (Figure 2).

### Challenges of AVE

In the endoscopic diagnosis of both NERD and EoE, AVE is time-consuming and requires substantial skill and knowledge to accurately interpret the findings. More recently, artificial intelligence (AI) was applied to the endoscopic diagnosis of EoE and has a significantly high sensitivity and specificity of over 90% [40]. AI will be helpful to resolve the problem of interpretation, especially for inexperienced endoscopists. We look forward to an ideal diagnostic endoscopy system using a combination of AVE and AI that should be simple, make examinations shorter, have high reproducibility, and be highly accurate in the diagnosis of NERD and EoE. 

## 4. Conclusions 

The use of AVE will improve the diagnostic accuracy of WLE in patients with NERD and EoE. Although AVE has several drawbacks, its improved accuracy may result in a reduced number of biopsies, especially in the diagnosis of EoE. Further large-scale randomized controlled trials are needed to investigate the diagnostic accuracy of AVE (including microscopic endoscopy and AI-assisted endoscopy) and to prove the positive effect of AVE on WLE in the diagnosis of NERD and EoE with a high evidence level. 

## Figures and Tables

**Figure 1 healthcare-10-02183-f001:**
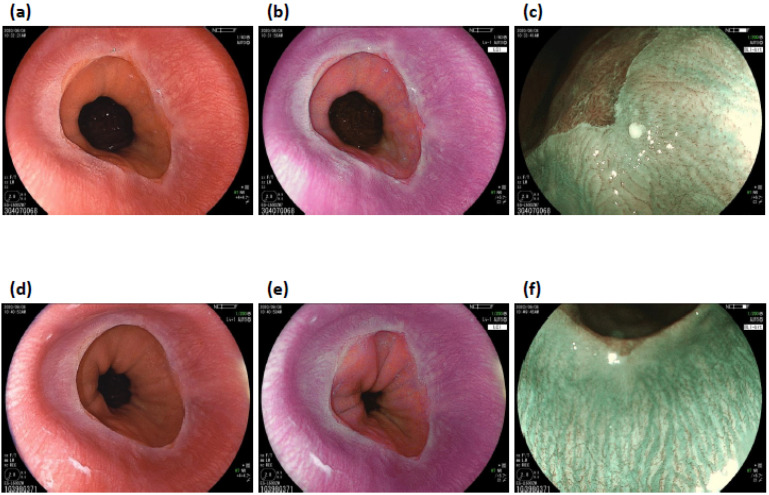
A 28-year-old male patient suffered from postprandial heartburn. The F-scale, which is a questionnaire scoring system for GERD, showed 27 points in this patient, which suggests reflux esophagitis since the result is higher than 8 points. WLE shows minimal changes in a white turbidity area with invisibility of palisade vessels along with slightly irregular z-line (**a**). LCI can visualize the minimal changes more clearly than WLE (**b**). BLI-ME demonstrates an increased number of dilated microvessels (IPCLs) in the white turbidity area (**c**). After taking a proton pump inhibitor for 4 weeks, the F-scale score decreased from 27 to 5 points. Color changes in white turbidity reduced on both WLE and LCI images ((**d**,**e**), respectively). BLI-ME shows that palisade vessels are visible and the severity of increased and dilated IPCLs reduces (**f**).

**Figure 2 healthcare-10-02183-f002:**
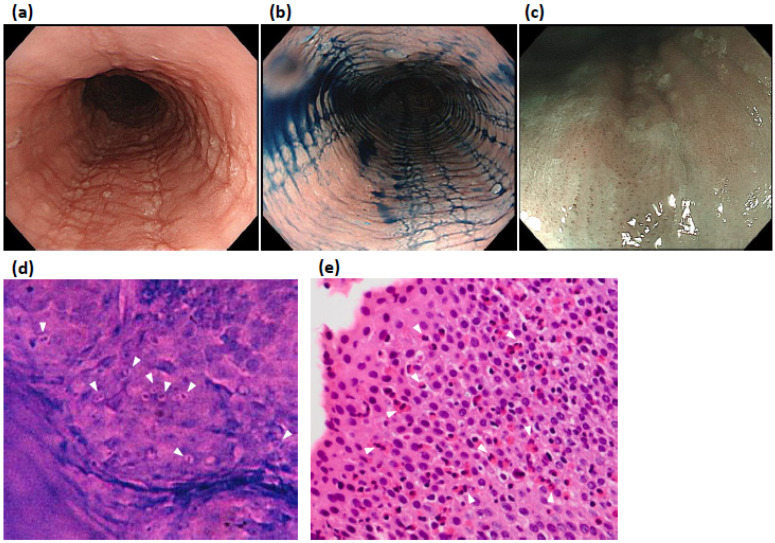
A 52-year-old male was referred to our university hospital because of heartburn and dysphagia that was refractory to proton pump inhibitors. WLE and imdigocarmine chromoendoscopy revealed linear furrows, exudates, and rings in the upper and middle esophagus (**a**,**b**). NBI-ME showed a beige-colored mucosa and dot-shaped IPCLs in the mucosal sites without visibility of cyan vessels (**c**). Endocytoscopy (magnification ×520) demonstrates multiple cells with purple bilobed nuclei with a nonstained periphery ((**d**), white arrowheads). This finding was suggestive of eosinophilic infiltration. Biopsy samples were collected at the site of endocytoscopic observation. The histology showed >50 eosinophils, several showing bilobed nuclei per high-power field ((**e**), white arrowheads showing representative bilobed nuclei of eosinophilic cells).

**Table 1 healthcare-10-02183-t001:** Diagnosis of Eosinophilic Esophagitis in Japan *****.

**Required items**
1. Esophageal symptoms, such as dysphagia.
2. ≥20 eosinophils/high power field on esophageal biopsy (multiple esophageal biopsies are recommended)
**Reference items**
1. Endoscopic findings of white plaque, linear furrows, and tracheal-like stenosis
2. CT or endoscopic ultrasound shows thickening of the esophageal walls
3. Increased presence of eosinophils in peripheral blood
4. Male
5. Unresponsiveness to PPI therapy and effectiveness to glucocorticoid

*****https://www.nanbyou.or.jp/entry/3935, accessed on 5 July 2022 (in Japanese).

**Table 2 healthcare-10-02183-t002:** Endoscopic reference score (EREFS) for eosinophilic esophagitis.

**Major features**
**EREFS-Inflammatory features**
**Edema** (also referred to as decreased vascular pattern, mucosal pallor)
Grade 0: absent (distinct vascularity present)
Grade 1: loss of clarity of vascular markings
**Exudates** (also referred to as white spots, plaques)
Grade 0: none
Grade 1: mild (lesions involving < 10% of the esophageal
surface area)
Grade 2: severe (lesions involving > 10% of the
esophageal surface area)
**Furrows** (also referred to as vertical lines, longitudinal furrows)
Grade 0: absent
Grade 1: present
**EREFS-Fibrotic features**
**Fixed rings** (also referred to as concentric rings, corrugated
esophagus, corrugated rings, ringed esophagus, trachealisation)
Grade 0: none
Grade 1: mild (subtle circumferential ridges)
Grade 2: moderate (distinct rings that do not impair passage of
standard diagnostic adult endoscope (outer diameter 8–9.5 mm)
Grade 3: severe (distinct rings that do not permit passage of
a diagnostic endoscope)
**Stricture**
Grade 0: absent
Grade 1: present
**Minor features**
Crepe paper esophagus (mucosal fragility or laceration upon
passage of diagnostic endoscope but not after esophageal dilatation)
Grade 0: absent
Grade 1: present

## Data Availability

Not applicable.

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
