# Peer review of "Advanced Endoscopy for Benign Esophageal Disease: A Review Focused on Non-Erosive Reflux Disease and Eosinophilic Esophagitis"

_healthcare, 2022, doi:10.3390/healthcare10112183_

Round 1

Reviewer 1 Report

I read your manuscript with great interests. I felt that the past important article is almost covered. I think your review is suitable of acceptance. However, I have a few questions. Please response my questions.

(minor comment)

1. What is ’SWE’ in line 61 (and line 67) in Page 2. You should spell out it.  

2. Please insert 'Figure1' and 'Figure2' into manuscript.

3. I think that images of upper-right and lower-right in Figure.1 are Blu LASER imaging (BLI), not NBI. Please correct in figure legend (NBI-ME→BLI-ME).  

4. There are some line break mistake, especially in figure legend. 

Author Response

(Response to reviewer 1)

Thank you very much for your valuable comments. We revised the manuscript according to your comments.

  1. We used “WLE” as the abbreviation of “standard white-light endoscopy” in the Introduction section and figure legends. We change “SWE” to “WLE” throughout the manuscript.
  2. We inserted Figure 1 in the page 4 and Figure 2 in the page 5. Endoscopic images of Figure 1 demonstrate the clinical utility of LCI and BLI in a case of NERD. Endoscopic images of Figure 2 show that NBI-ME was helpful in the diagnosis of EoE and endocytoscopy could visualize eosinophils infiltrating into the squamous epithelium directly.
  3. We changed “narrow band imaging magnification endoscopy (NBI-ME)” to “blue laser imaging magnification endoscopy (BLI-ME)”.
  4. We corrected the line break mistakes in legends of Figure 1 and Figure 2.

Reviewer 2 Report

To the author

This review article covered the current status of the advanced endoscopy for the diagnosis of benign esophageal disease. I carefully read this reviewed article and I could understand the current situation of this field.

I almost agree with this manuscript, however, there are some concerns before publication.

1.       Author explained the current situation of advanced endoscopy for the diagnosis of benign esophageal disease in this review article. I almost agree with the content of this manuscript, however, author should explain the method how to select the article which author selected in this manuscript.

2.       Author discussed the benign esophageal disease especially NERD and EoE in this manuscript. Author should explain the reason why author selected these two conditions in this manuscript.

3.       Author discussed the advantage and disadvantage of advanced endoscopy compared to the white light imaging in this manuscript. Author should discuss the merit and demerit between each advanced endoscopy.

4.       Author wrote “Biopsy of the esophageal mucosa showing >15 eosinophil” in Table1. Author should confirm >15 is correct or not.

Author Response

(Response to reviewer 2)

We appreciate for your helpful comments. We revised the manuscript according to your comments.

  1. We explored and selected previous studies that will be suitable for this review article using PubMed with the keywords “non-erosive reflux disease (NERD)”, “eosinophilic esophagitis (EoE)”, “endoscopic diagnosis”, “image-enhanced endoscopy”, “magnification endoscopy, “and “advanced endoscopy”. We added the explanation in page 2 in the end of Introduction
  2. As mentioned in the Abstract, we focused on non-erosive esophagogastric reflux disease (NERD) and eosinophilic esophagitis (EoE) which sometimes have no abnormal finding under standard white light endoscopy alone. Additionally, NERD and EoE are increasing prevalence rates recently as mentioned in the Introduction. We added the sentence in the Introduction section as follows: Herein, we present a review of the clinical significance of the use of AVE, particularly NBI, in the endoscopic diagnosis of benign esophageal disease, non-erosive reflux disease (NERD), and eosinophilic esophagitis (EoE), which have recently had an in-crease in prevalence. Additionally, it is still difficult to diagnose NERD and EoE endoscopically because there are sometimes no clear findings with WLE alone, as mentioned below.
  3. To response comments of you and reviewer 4, we added a new paragraph, namely “Challenges of AVE” before conclusion and describe demerits of advanced endoscopy in the endoscopic diagnosis of both NERD and EoE as follows: Challenges of AVE In the endoscopic diagnosis of both NERD and EoE, AVE is time-consuming and requires substantial skill and knowledge to accurately interpret the findings. More recently, artificial intelligence (AI) has been applied to the endoscopic diagnosis of EoE and has a significantly high sensitivity and specificity of over 90% [40]. AI will be helpful to resolve the problem of interpretation, especially for inexperienced endoscopists. We look forward to an ideal diagnostic endoscopy system using a combination of AVE and AI that should be simple, make examinations shorter, have high reproducibility, and be highly accurate in the diagnosis of NERD and EoE.
  4. We apologize for the incorrect description. We confirmed the homepage of “Japan intractable diseases information center” (https://www.nanbyou.or.jp/entry/3935 (in Japanese)) and revised the Table 1 as follows:

Required items

 1.  Esophageal symptoms, such as dysphagia.

 2.  20 eosinophils/high power field on esophageal biopsy (multiple esophageal biopsies are recommended)

Reference items

 1.  Endoscopy of the esophagus indicates white plaque, linear furrows, and tracheal-like stenosis

 2.  CT or endoscopic ultrasound shows thickening of the esophageal walls

 3.  Increased presence of eosinophils in peripheral blood

 4.  Male  

 5.  Unresponsiveness to proton pump inhibitor therapy and effectiveness to glucocorticoid

Reviewer 3 Report

I reviewed with interest the article entitled “Advanced endoscopy for benign esophageal disease: A review focused on non-erosive esophagogastric reflux disease and eosinophilic esophagitis”. This review is interesting and well organized. I’m sure that our readers will find it of great interest. I have only some minor comments to be addressed as follows:

1.     Page2, Line47. “80% of patients with symptoms of GERD also experience NERD at a minimum of twice per week” What does this sentence mean? Does it mean “80% of NERD patients experience GERD symptoms at a minimum of twice per week”? And please indicate the reference of this sentence.

2.     Page2, Line 61 and 67. What does SWE stand for?

3.     Please cite figures in the text in appropriate parts.

4.     Page3, Line 78. The concept of “Grade M” is not accepted or recognized worldwide. It may require a more detailed explanation (maybe at Page2, Line 54?) because it is mentioned several times after this.

5.     Page 4, Line 144. Does this sentence need to cite the guideline as a reference?

Author Response

(Response to reviewer 3)

We appreciate for your valuable comments. We revised the manuscript according to your comments.

  1. The sentence that you suggested is correct. We revised the sentence and site the reference mentioned below.80% of patients with NERD experience GERD symptoms at least twice per week [7]

7. Japan Gastroenterological Endoscopy Society, ed., Gastroenterological  endoscopy handbook (2nd edition). Tokyo: Nihon Medical Center, 2017, p.185 (in Japanese).

  1. We used “WLE” as the abbreviation of “standard white-light endoscopy” in the Introduction section and figure legends. We change “SWE” to “WLE” throughout the manuscript.
  2. We inserted Figure 1 in the page 4 and Figure 2 in the page 5 before the new paragraph of “Challenges of AVE”. Endoscopic images of Figure 1 demonstrate the clinical utility of LCI and BLI in a case of NERD. Endoscopic images of Figure 2 show that NBI-ME was helpful in the diagnosis of EoE and endocytoscopy could visualize eosinophils infiltrating into the squamous epithelium directly.
  3. Grade M is a revised Los Angeles classification. Endoscopic findings of Grade M include minimal mucosal changes of white turbidity or erythema without erosion in patients with GERD that correspond to NERD. We added the explanation in page 3.
  4. We added the reference mentioned below in the reference list.                 27. The Japanese society of Gastroenterology, ed., Evidence-based clinical practice guideline for gastroesophageal reflux disease (GERD) 2021 (3rd edition). Tokyo: Nankodo, 2021, p.37 (in Japanese)

Author Response

(Response to Reviewer 4)

Thank you very much for your thoughtful instruction and valuable comments.

  1. - Line 125: “FICE, a digital technology similar to NBI” is a vague description, I suggest

providing a more in-depth explanation in the additional paragraph concerning the endoscopy methods.

(Response) We added the explanation as follow: FICE, a digital technology similar to NBI that allows for enhanced visibility of micro-structures of vessels and mucosal structures

  1. - It is enough to describe and mention the EREFS once in the text and not twice (line 160 and 182)

(Response) We deleted the words indicating EREFS (line 182).

  1. - Figures are not mentioned in the text. Please include the reference to the figures in the manuscript texts.

(Response) We inserted Figure 1 in the page 4 and Figure 2 in the page 5. Endoscopic images of Figure 1 demonstrate the clinical utility of LCI and BLI in a case of NERD. Endoscopic images of Figure 2 show that NBI-ME was helpful in the diagnosis of EoE and endocytoscopy could visualize eosinophils infiltrating into the squamous epithelium directly.

  1. - The Figure legends resemble case reports. I suggest rewriting the legends into a concise form

(Response) We revised the figure legends as concise as possible.

  1. - “Increased IPCLs” (in the Figure 1 legend and line 98)-do you mean “increased number of IPCLs” or something else?

(Response) According to your suggestion, we changed “Increased IPCLs” to “increased number of IPCLs” throughout the manuscript.

  1. - The Conclusions need to be improved. AI should be mentioned in the text as one of

the advanced methods and not in the conclusions. The difficulty of performing the AVE is not considered in the text, so it cannot be concluded that the techniques are not easy to perform and interpret (unless publications on the learning curve and technical difficulties are included in the review). I would recommend mentioning which methods nowadays have already proven useful and which require further investigation. Also, it seems important to mention the clinical usefulness of the AVE (e.g. the need for less biopsies, more accurate biopsies etc).

(Response) According to your suggestion, we made a new paragraph of “Challenges of AVE” to describe the drawbacks of AVE and AI potential for resolving one of the drawbacks.

  1. - English revision would be useful, the text is mostly well written but with some errors

which occasionally cause confusion

(Response) We have already corrected this review article by a proofreader, but received proofreading again. Please refer to the attached file.

Round 2

Reviewer 4 Report

Dear Authors, thank you for revision of the Manuscript. I find the corrections appropriate.